# Longitudinal Profiles of Anti-Platelet Factor 4 Antibodies in Thai People Who Received ChAdOx1 nCoV-19 Vaccination

**DOI:** 10.3390/vaccines11030692

**Published:** 2023-03-17

**Authors:** Nonthakorn Hantrakun, Peampost Sinsakolwat, Adisak Tantiworawit, Ekarat Rattarittamrong, Thanawat Rattanathammethee, Sasinee Hantrakool, Pokpong Piriyakhuntorn, Teerachat Punnachet, Piangrawee Niprapan, Ornkamon Wongtagan, Romanee Chaiwarith, Lalita Norasetthada, Chatree Chai-Adisaksopha

**Affiliations:** 1Division of Hematology, Department of Internal Medicine, Faculty of Medicine, Chiang Mai University, Chiang Mai 50200, Thailand; 2Division of Infectious Diseases and Tropical Medicine, Department of Internal Medicine, Faculty of Medicine, Chiang Mai University, Chiang Mai 50200, Thailand

**Keywords:** COVID-19 vaccine, vaccine-induced thrombocytopenia and thrombosis, VITT, Asian

## Abstract

Anti-platelet factor 4 (anti-PF4) antibodies were identified as pathogenic antibodies for vaccine-induced immune thrombocytopenia and thrombosis (VITT) in subjects receiving ChAdOx1 nCoV-19 vaccinations. We performed a prospective cohort study to determine the prevalence of anti-PF4 and the effect of the ChAdOx1 nCoV-19 vaccine on anti-PF4 in healthy Thai subjects. Anti-PF4 antibodies were measured before and four weeks after receiving the first vaccination. Participants with detectable antibodies were scheduled for repeat anti-PF4 analysis at 12 weeks after the second vaccination. Of 396 participants, ten participants (2.53%; 95% confidence interval [CI], 1.22–4.59) were positive for anti-PF4 before receiving vaccinations. Twelve people (3.03%; 95% CI, 1.58–5.23) had detectable anti-PF4 after the first vaccination. There was no difference in the optical density (OD) values of anti-PF4 antibodies when comparisons were made between pre-vaccination and four weeks after the first vaccination (*p* = 0.0779). There was also no significant difference in OD values in participants with detectable antibodies. No subjects experienced thrombotic complications. Pain at the injection site was associated with an increased risk of being anti-PF4 positive at an odds ratio of 3.44 (95% CI, 1.06–11.18). To conclude, the prevalence of anti-PF4 was low in Thais and did not significantly change over time.

## 1. Introduction

The outbreak of the severe acute respiratory syndrome coronavirus 2 (SARS-CoV-2) pandemic, which then led to the coronavirus disease (COVID-19), has been declared a global health emergency. As of May 2022, there were over 10 million estimated deaths from COVID-19, as has been reported by the Institute for Health Metrics and Evaluation (IHME) [1]. Accordingly, increases in incidences of COVID-19, and associated mortality rates, have been observed worldwide [1].

While a number of new therapeutic approaches have been developed to improve the outcomes of the disease, one of the major medical interventions aimed at lowering the mortality and incidence rates of COVID-19 is the SARS-CoV-2 vaccine [2]. Several different types of vaccines have currently been made available [3,4]. The ChAdOx1 nCoV-19 adenoviral vector-based vaccine (Oxford/AstraZeneca) demonstrated efficacy in preventing symptomatic and severe COVID-19 in older adults [5]. In Thailand, a two-dose regimen of the ChAdOx1 nCoV-19 vaccine has exhibited protective effects and reduced the risk of the severity of the disease [6]. Any adverse effects following immunization with the ChAdOx1 nCoV-19 vaccine were found to be mild and tolerable [7,8]. However, there have been incidences of patients experiencing thrombosis, thrombocytopenia, and elevated D-dimer levels after being vaccinated with the ChAdOx1 nCoV-19 vaccine. Remarkably, the serious adverse effect associated with the ChAdOx1 nCoV-19 vaccine has been identified as an increased risk of thrombosis [9,10]. This syndrome has come to be known as vaccine-induced immune thrombocytopenia and thrombosis (VITT) [11].

VITT is a rare but potentially fatal complication characterized by unusual site thrombosis followed by thrombocytopenia after receiving the adenoviral vector-based vaccination for COVID-19. Heparin-induced thrombocytopenia antibodies or anti-platelet factor 4 (anti-PF4) antibodies have been identified as pathogenic antibodies by enzyme-linked immunosorbent assays (ELISAs), which have also detected anti-PF4/polyanion immunoglobulin G (IgG) antibodies [12]. Thrombocytopenia (platelet counts less than 150 × 10^9^/L) and disproportionately elevated D-dimer levels, particularly those that are greater than four times the upper normal limit, have also been indicated in the laboratory findings of VITT patients [11,13].

The immune complex of anti-PF4 antibodies is the key pathogenic antibody found in VITT. Anti-PF4 antibodies activate both platelets and neutrophils through FcγRIIa receptors [14]. Activated neutrophils release neutrophil extracellular traps (NETs) via a multistep process referred to as NETosis [15]. Both platelet activation and NETosis can lead to thrombocytopenia, massive thrombin generation, and thrombus formation. Thrombotic events often involve multiple vascular beds, as well as both arterial and venous circulations.

The diagnostic criteria for VITT comprise five domains: onset of symptoms after vaccination; presence of thrombosis; thrombocytopenia; D-dimer levels; and positive anti-PF4 antibodies [11]. Patients who do not meet all five criteria are considered to have either probable, possible, or unlikely VITT based on the proposed diagnostic criteria [11]. Anti-PF4 antibodies can be detected in healthy individuals [16], whereas patients who are suspected to have VITT will exhibit clinical findings and laboratory results compatible with VITT.

The clinical features of VITT commonly begin 5–10 days after patients received the ChAdOx1 nCoV-19 vaccine. Among 294 VITT patients, the median time for the initial symptoms was 14 days (range 5 to 48 days) [11]. The most common site of thrombosis was cerebral venous sinus thrombosis, which accounted for 50% of patients who had definite or probable VITT [11]. Apart from cerebral venous sinus thrombosis, other sites of venous thrombosis have also been reported including deep vein thrombosis of the leg, pulmonary embolism, splanchnic vein thrombosis, or others. Arterial thrombosis was observed in 21% of VITT patients. The clinical presentation of arterial thrombosis included myocardial infarction, cerebrovascular disease, or acute limb ischemia [11]. Secondarily, bleeding resulting from anticoagulant therapy, was observed in some patients, particularly in those who had cerebral venous sinus thrombosis [9,10]. VITT is associated with high incidences of mortality, which was reported in 22% of subjects after the initial diagnosis [11]. Due to the high degree of severity of the disease, VITT should be suspected in individuals who had recently received the ChAdOx1 nCoV-19 vaccination and developed clinical thrombo-hemorrhagic features.

The degree of incidence of VITT has ranged from one case per 26,500 (Norway) to one case per 127,300 (Australia) upon receiving the first dose of the ChAdOx1 nCoV-19 vaccine [17]. However, comprehensive data on VITT in Asian countries has been limited. Previous Thai data have indicated that anti-PF4 antibody frequencies were 2.3–3.1% after patients received the first dose of the ChAdOx1 nCoV-19 vaccine [16,18], whereas the degree of incidence of VITT was estimated to be one in three million [19].

Because the prevalence of anti-PF4 antibodies and VITT has been extremely rare in patients who had received the ChAdOx1 nCoV-19 vaccine, the long-term effects of the ChAdOx1 nCoV-19 vaccine on any dynamic changes in levels of anti-PF4 antibodies have not been well demonstrated. Repeated doses of the vaccine are required to achieve sufficient protection against COVID-19 [20]. However, data related to safety issues among Asian patients with detectable anti-PF4 antibodies, who had also received a second dose or booster dose of the ChAdOx1 nCoV-19 vaccine, have been limited.

This study aimed to evaluate the presence of anti-PF4 antibodies, signs and symptoms of thromboembolic events, the degree of incidence of VITT, and the effect of the ChAdOx1 nCoV-19 vaccine on anti-PF4 antibodies among individuals who had received the ChAdOx1 nCoV-19 vaccine, and on those who had received a longitudinal follow-up.

## 2. Materials and Methods

### 2.1. Study Design and Participants

This was a prospective cohort study conducted at a university-based hospital located in Thailand from June of 2021 to January of 2022. Participants aged ≥ 18 years who were committed to receiving the ChAdOx1 nCoV-19 vaccine were enrolled in this study. Participants with a history of recent heparin or low-molecular weight heparin-use within three months of the study, those undergoing current anticoagulant use, or those in an immunocompromised state due to an underlying disease or those who had received immunosuppressive drugs, those with a recent history of major surgery within three months of the study, or those who had a history of platelets < 150 × 10^9^/L within three months of the study were all excluded from participating. Informed consent was obtained from all participants. This study was conducted following the Declaration of Helsinki and approved by the Research Ethics Committee, Faculty of Medicine, Chiang Mai University [Approval No. 160/2021].

### 2.2. Participant Recruitment and Trial Intervention

We recruited participants at a university-based hospital vaccine service center. All participants who planned to receive the ChAdOx1 nCoV-19 vaccine with a typical interval of 12 weeks between doses were invited. Participants were screened for eligibility.

Demographic data were collected, including age, sex, underlying diseases, and current medications. An initial blood sample was collected to measure anti-PF4 antibodies before vaccination. All participants’ adverse reactions were assessed after being given the vaccination. After being given the vaccination, participants were asked to self-report solicited adverse events using a paper-based questionnaire. Solicited adverse events in the questionnaire were classified as local and systemic adverse events including fever, headache, asthenia, hypoesthesia, arthralgia, pain at the injection site, dyspnea, myalgia, rash, fatigue, and pruritus. All participants were advised to be aware of symptoms related to thrombosis and unsolicited adverse events during the study period. A hospital vaccination service was available for consultation if participants suspected unsolicited complications after receiving the ChAdOx1 nCoV-19 vaccine. Participants were evaluated for thrombotic complications four weeks after receiving the first dose of the vaccine. A second blood sample was then collected to test for anti-PF4 antibodies at this time.

All participants were also evaluated for thrombotic complications four weeks after receiving the second dose of the ChAdOx1 nCoV-19 vaccine. A Third blood sample was collected to detect anti-PF4 antibodies. D-dimer testing was then initiated and a complete blood count (CBC) was administered in participants exhibiting detectable anti-PF4 after the first vaccination.

### 2.3. Laboratory Measurements

#### 2.3.1. Anti-PF4/Heparin Antibodies

Blood samples of all participants were collected twice to measure anti-PF4 antibodies: before the first dose of the vaccine and four weeks after the first dose of the vaccine. Participants who had detectable anti-PF4 antibodies after receiving the first dose of the vaccine were scheduled for the third blood sample after 12 weeks of receiving the second dose of the vaccine in order to again test for the presence of anti-PF4 antibodies. Blood samples were collected in a citrated tube as appropriate. Specimens were processed within two hours after collection. Platelet-poor plasma was stored at −70 °C until it was tested. Anti-PF4/heparin antibodies of the IgG isotype were determined using the ZYMUTEST HIA, IgG ELISA (Hyphen BioMed, Neuville-sur-Oise, France). We defined the negative cutoff optical density (OD) value as ≤0.3. Moreover, OD values of >0.3 to 0.5, and >0.5 were assigned as weakly positive, and positive, respectively.

#### 2.3.2. D-Dimer

All participants with detectable anti-PF4 antibodies had agreed to receive D-dimer testing at the time that they were scheduled for their third anti-PF4 antibody testing. Blood samples were collected in a citrated tube. Specimens were tested within two hours after being collected. D-dimer was measured by particle-enhanced immunoturbidimetric assay on the Sysmex^®^ CS-2500 system (Sysmex, Kobe, Japan) using INNOVANCE^®^ D-dimer reagent.

#### 2.3.3. Complete Blood Count (CBC)

All participants with detectable anti-PF4 antibodies received CBC testing at the time that they were scheduled for their third anti-PF4 antibody testing. Blood samples were collected in an EDTA tube. CBC was assessed using the automated Sysmex^®^ XN-9000 hematology analyzer (Sysmex, Kobe, Japan).

### 2.4. Statistical Analysis

Participants who had received complete laboratory results from both pre- and post-vaccination blood samples after being given the first dose of the ChAdOx1 nCoV-19 vaccine were analyzed using a per-protocol analysis. Participants who did not attend a follow-up session or who had incomplete results with regard to both pre- and post-vaccination anti-PF4 antibodies were excluded from the final analysis.

Baseline characteristics were described using descriptive statistics. Mean values with corresponding standard deviation (SD) or median values with interquartile range (IQR) were used to summarize the data as appropriate. Univariable comparisons were performed using the Mann–Whitney U test or Student’s *t*-test for continuous variables, and Fisher’s exact probability test for categorized variables. We performed a pairwise Student’s *t*-test or the Mann–Whitney U test to determine any difference in the average OD values. The factors that affected the positivity of the anti-PF4 antibodies were analyzed by univariable logistic regression analysis. All data in this study were analyzed using Stata 16 (StataCorp, College Station, TX, USA). *p*-values < 0.05 were considered statistically significant.

## 3. Results

A total of 396 participants were selected for the analysis (Figure 1). Demographic and the clinical characteristics of all participants are shown in Table 1. The median age (range) of participants was 50 (18–87) years. Female subjects made up 65.91% of the study population. The most common comorbid disease was hypertension (32.32%), followed by dyslipidemia (25.51%), diabetes mellitus (12.12%), a history of coronary artery disease (3.03%), and a history of experiencing ischemic stroke (1.52%). There were 16 (4.04%) participants who were currently receiving treatment with aspirin.

Before receiving the vaccination, ten participants (2.53%; 95% confidence interval [CI], 1.22–4.59) had positive anti-PF4 antibodies. These subjects were then separated as follows: two for the weakly positive group and eight for the positive group.

At four weeks after administration of the first dose of the ChAdOx1 nCoV-19 vaccine, twelve participants had detectable anti-PF4 antibodies (two new cases: one for the weakly positive group, one for the positive group) given a degree of prevalence of 3.03% (95% CI, 1.58–5.23). No differences were observed in the median OD values of anti-PF4 antibodies in comparisons made between pre-vaccination subjects and four weeks after subjects had received their first dose of the ChAdOx1 nCoV-19 vaccine (0.029 (0.01–0.064) [median (IQR)] for pre-vaccination subjects and 0.032 (0.009–0.068) for four weeks after those subjects had received their first dose of the vaccine; *p* = 0.0779) (Figure 2).

### 3.1. Adverse Reactions after Receiving the Vaccination

Common adverse reactions after receiving the first dose of the ChAdOx1 nCoV-19 vaccine were fever (29.04%), followed by pain at the injection site (17.93%), fatigue (15.91%), headaches (9.34%), and arthralgia (6.57%). Other note-worthy adverse reactions, including asthenia, hypoesthesia, dyspnea (transient), myalgia, rash, and pruritus, are presented in Table 2.

### 3.2. Participants with Anti-PF4 Positive

Among subjects who were anti-PF4 positive, the results of complete blood counts and d-dimer tests are presented in Table 3. The mean (± SD) hemoglobin level was recorded at 13.46 ± 1.17 g/dL, and the mean (± SD) platelet count was recorded at 296.50 ± 65.25 × 10^9^/L. No participant had a platelet count less than 150 × 10^9^/L. The mean D-dimer level was 288.58 ± 163.37 ng/mL. There was only one participant (8.33%) with a D-dimer result greater than 500 ng/mL (738 ng/mL).

All subjects who had detectable anti-PF4 antibodies after receiving the first dose of the vaccine were scheduled for the third blood sample. At 12 weeks after the second vaccination was administered, eleven participants had detectable anti-PF4 antibodies. Furthermore, all eleven participants had either weakly positive or positive results after receiving the first vaccination. One participant had seroconversion results switching from weakly positive to undetectable antibodies.

There were three participants who exhibited a dynamic change in levels of anti-PF4 antibodies. One participant had weakly positive anti-PF4 antibodies before and after receiving the first vaccination dose; however, anti-PF4 antibodies were undetectable at 12 weeks after receiving the second vaccination dose. One participant had undetectable anti-PF4 antibodies before receiving the first dose of the vaccine, but the antibodies turned weakly positive after the first vaccination and remained weakly positive after receiving the second dose of the vaccine. One participant had undetectable anti-PF4 antibodies before receiving the first dose of the vaccine, but the antibodies turned positive after they received the first vaccination dose and remained positive after the second dose of the vaccine.

The median OD values (IQR) in participants with detectable anti-PF4 antibodies were 0.760 (0.408–0.982) when measured during the pre-vaccination period, 0.769 (0.549–0.991) when measured four weeks after the first vaccination, and 0.942 (0.539–1.375) when measured 12 weeks after the second dose of the ChAdOx1 nCoV-19 vaccine, respectively. There were also no significant differences in OD values of the anti-PF antibodies when comparisons were made among these three periods (*p* = 0.2427 for pre-vaccination vs. four weeks after the first dose, *p* = 0.0564 for pre-vaccination vs. 12 weeks after the second dose, and *p* = 0.0544 for four weeks after the first dose vs. 12 weeks after the second dose) (Figure 3).

### 3.3. Univariable Analysis of the Factors Associated with Positivity of Anti-PF4 Antibodies

Table 4 demonstrates a univariable analysis of the associated factors and the occurrence of anti-PF4 antibodies. Pain at the injection site was associated with an increased risk of anti-PF4 antibodies positive at an odds ratio of 3.44 (95% CI, 1.06–11.18; *p* = 0.040). Other factors were not significantly associated with the positivity of anti-PF4 antibodies.

### 3.4. Thrombotic Complications

Four weeks after receiving the first and second doses of the ChAdOx1 nCoV-19 vaccine, no participant developed signs and symptoms of thrombotic complications. Bleeding complications were also not observed in our study.

## 4. Discussion

Our findings suggest that the degree of prevalence of anti-PF4 antibodies after administering the ChAdOx1 nCoV-19 vaccine was low (3.03%). Moreover, we observed that 2.53% of people had preexisting anti-PF4 antibodies. According to the results of previously conducted Thai studies, the prospective cohort data obtained from Thailand demonstrated that the prevalence of anti-PF4 antibodies was 2.3–3.1% after receiving the first dose of the ChAdOx1 nCoV-19 vaccination [16,18]. Regarding the results of studies conducted in various Western countries, the prevalence of anti-PF4 antibodies in subjects after receiving the first dose of the ChAdOx1 nCoV-19 vaccination was also low, and the OD values were mainly low, primarily between 0.5 and 1. In Norwegian and German studies, the degrees of frequency of anti-PF4 antibodies were reported to be 1.2% (six from 492 participants, using LIFECODES PF4 IgG ELISA with a cutoff OD value ≥ 0.4) [21] and 8% (11 from 138 participants using in-house IgG-specific PF4/polyanion enzyme immunoassays with a cutoff OD value ≥ 0.5) [22]. Remarkably, none of these participants had a low platelet count [21]. Any variations in the percentages among the studies may be attributed to the sensitivities of the laboratory assays used [23]. Importantly, none of the detectable anti-PF4 antibodies exhibited the functional ability to activate the platelet activation assay in all cohorts, whereas no subjects developed symptomatic VITT [16,18,21,22]. All these findings indicate that pathogenic platelet-activating antibodies did not occur commonly following administration of the vaccination, whereas the positivity of anti-PF4 antibodies in asymptomatic participants lacked any association with symptomatic VITT.

There are discrepancies among incidences of VITT and the frequency of anti-PF4 antibodies. Since there is no such standard measurement for anti-PF4 antibodies and the cut-off point for antibody positivity, it is difficult to compare the results of studies investigating the frequency of occurrence of anti-PF4 antibodies. Platton et al. evaluated the laboratory results for anti-PF4 antibodies among patients with suspected VITT through the use of IgG-specific ELISAs, polyspecific ELISAs, and rapid assays. The main findings indicate that rapid assays (HemosIL HIT-Ab, HemosIL AcuStar HIT-IgG, STic Expert assays, and Diamed PaGIA gel) were associated with a poor degree of sensitivity for VITT in comparison to the results of the ELISAs [23]. The International Society on Thrombosis and Haemostasis Scientific and Standardization Committee (ISTH-SSC) on platelet immunology have recommended an antigen-binding assay (ELISAs) for testing anti-PF4 antibodies. Rapid immunoassay (RIA) and chemiluminescence immunoassay (CLIA) should be avoided due to the concern that these tests may reveal false-negative results. In the present study, we elected to use an IgG-specific ELISA (ZYMUTEST HIA, IgG ELISA (Hyphen BioMed, Neuville-sur-Oise, France) according to the ISTH-SSC recommendation. Although, a cut-off point was not defined, we employed the cut-off point that was recommended by the manufacturer.

Anti-PF4 antibodies may naturally occur in the general heparin-naïve population [24]. A seroprevalence study using samples obtained from people who donated blood to the American Red Cross indicated that the prevalence of anti-PF4/heparin antibodies was around 4.3–6.6% [24]. Our study demonstrated that 2.53% of participants had detectable anti-PF4 antibodies before receiving the first dose of the ChAdOx1 nCoV-19 vaccine. We further analyzed the differences in the OD values before and four weeks after subjects received the first dose of the ChAdOx1 nCoV-19 vaccine. The results indicated that there was no significant difference in the OD values of anti-PF4 antibodies when comparisons were made between pre-vaccination subjects and those who had received the first dose of the ChAdOx1 nCoV-19 vaccine after a four-week period (*p* = 0.0779) (Figure 2). Therefore, detection of the anti-PF4 antibodies should be carefully interpreted along with the relevant correlated clinical manifestations.

Supporting evidence for the VITT diagnosis involved D-dimer levels [10,11,13,25]. Subjects were unlikely to be diagnosed with VITT when D-dimer levels were lower than 2000 fibrinogen equivalent units (FEUs) or lower than 1000 ng/mL [11]. The findings in our study showed that among participants with detectable anti-PF4 antibodies, there was only one person with a D-dimer level greater than 500 ng/mL. Remarkably, this participant had no signs and symptoms of thrombotic diseases. There were numerous potential factors influencing the elevation in D-dimer values such as transient infection or inflammation, injuries, hypertension [26], or systemic sclerosis [27]. Elevated D-dimer levels could also be observed in healthy people, especially in members of the elderly population [28]. Thrombocytopenia was also linked to VITT, particularly when a platelet count < 150 × 10^9^/L was recorded [11]. However, in our study, there was also no participant who developed thrombocytopenia with a platelet count < 150 × 10^9^/L. Therefore, these findings suggest that routine D-dimer and CBC testing after administration of the ChAdOx1 nCoV-19 vaccination may, in fact, be an unnecessary investigation in people with no clinical suspicion of thrombotic diseases.

We have demonstrated the effect of the ChAdOx1 nCoV-19 vaccine on anti-PF4 antibodies by measuring them three times in participants with detectable anti-PF4 antibodies after receiving the first dose of the vaccination. Among the three measurements, there were no significant differences in OD values with regard to anti-PF4 antibodies, which was in line with the increased trends of the OD values. As was similar to the outcomes of our study, a recent study found no differences in anti-PF4 antibody generation and no evidence of altered anti-PF4 antibody functionality in people with preexisting nonpathogenic anti-PF4 antibodies [29]. To date, the risk of COVID-19 remains a serious public health problem, whereas vaccination against SARS-CoV-2 can help protect against the disease. Our results provide support for the safety of administering the ChAdOx1 nCoV-19 vaccination as a second or booster dose, if necessary, even in asymptomatic people with previously positive anti-PF4 antibodies.

In our study, a significant risk factor for the development of anti-PF4 antibodies was pain at the injection site. To date, there has been no research on the potential of a direct relationship between experiencing pain during administration of the vaccine and anti-PF4 antibody formation. Likewise, risk factors for VITT remain unknown. Females and younger individuals were reported to be the majority of subjects with VITT in the initial case series [9,11]. This association has been considered controversial since the demographics of the first group of individuals who received the ChAdOx1 nCoV-19 vaccine were young female health-care workers. However, some research studies have formulated hypotheses about skin inflammation in subjects after receiving the vaccination indicating proinflammatory reactions to pathologic anti-PF4 antibody formation [30]. The study of any potential association between clinical and anti-PF4 antibody production should be further investigated.

The strength of our study was that we took blood samples from all participants before they received the first dose of the vaccination. As a result, we determined that some participants already had anti-PF4 antibodies. We also followed participants longitudinally and observed the natural course of anti-PF4 antibodies in participants who received the ChAdOx1 nCoV-19 vaccine. The dropout rate was low in our study. Moreover, we focused on participants who had positive results for anti-PF4 antibodies in terms of investigations of the laboratory and clinical features of VITT. We have acknowledged some limitations to our study. First, this was a single-center study which was limited to the Thai population. Therefore, the findings of this study may be limited to a generalizability to Asians. Second, we measured CBC and D-dimer levels at the time that participants were scheduled for their third anti-PF4 antibody testing (12 weeks after receiving the second dose of the ChAdOx1 nCoV-19 vaccine). Consequently, the results may not truly represent the effect of the vaccine on any hematologic parameters. Third, no participant in our study switched from the ChAdOx1 nCoV-19 vaccine to mRNA vaccine. Therefore, we did not produce any data with regard to anti-PF4 antibody profiles for those who may have switched vaccinations. Lastly, VITT is a rare complication of the ChAdOx1 nCoV-19 vaccine. This study did not demonstrate the nature of anti-PF4 antibodies in patients who had probable or definite VITT.

## 5. Conclusions

The frequency of anti-PF4 antibodies in subjects after they received the first dose of the ChAdOx1 nCoV-19 vaccination was low in this study. Accordingly, the second dose of ChAdOx1 nCoV-19 vaccine did not facilitate anti-PF4 antibody production, so it was determined to be safe for those subjects to receive the ChAdOx1 nCoV-19 vaccination as a second dose or booster dose. This was true even among asymptomatic people with previously positive anti-PF4 antibodies. To conclude, the prevalence of anti-PF4 antibodies was low in Thai people and did not significantly change over time.

## Figures and Tables

**Figure 1 vaccines-11-00692-f001:**
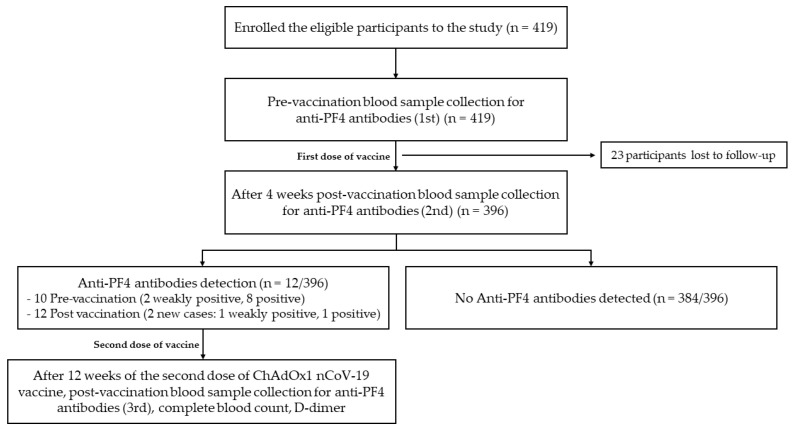
Cohort flow diagram of the study.

**Figure 2 vaccines-11-00692-f002:**
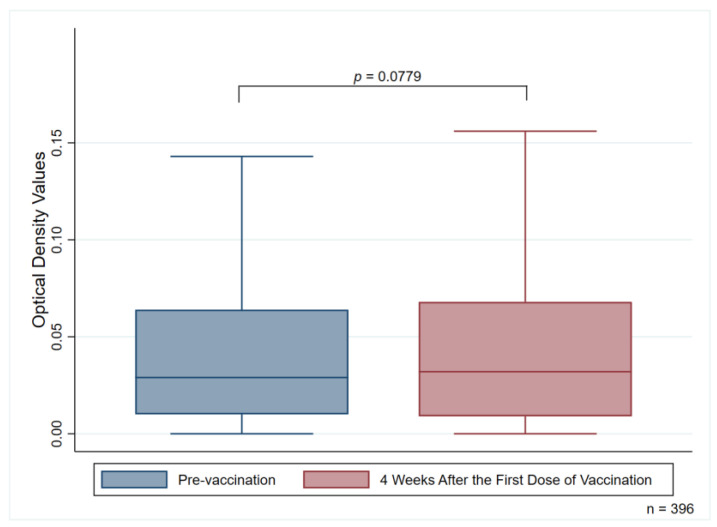
Median optical density values of anti-PF4 antibodies measuring pre-vaccination and four weeks after the first vaccination dose.

**Figure 3 vaccines-11-00692-f003:**
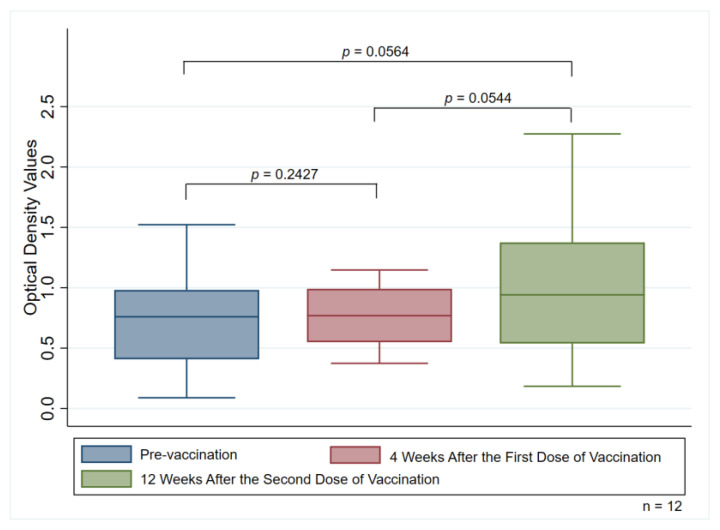
Median optical density values of anti-PF4 antibodies measuring pre-vaccination group, four weeks after the first dose, and 12 weeks after receiving the second dose of the vaccination among the 12 participants with detectable anti-PF4 antibodies.

**Table 1 vaccines-11-00692-t001:** Demographic data and clinical characteristics of participants.

Characteristics	All Participants (n = 396)
Age (year) (median (range))	50 (18–87)
Sex (female) (no. (%))	261 (65.91)
Comorbidities (no. (%))	
Hypertension	128 (32.32)
Dyslipidemia	101 (25.51)
Diabetes mellitus	48 (12.12)
History of coronary artery disease	12 (3.03)
History of ischemic stroke	6 (1.52)
Concomitant medication [no. (%)]	
Anti-hypertensive drug	128 (32.32)
Anti-glycemic drug	43 (10.86)
Aspirin	16 (4.04)

**Table 2 vaccines-11-00692-t002:** Symptoms after receiving the first dose of the ChAdOx1 nCoV-19 vaccination.

Characteristics [No. (%)]	All Participants (n = 396)
Fever	115 (29.04)
Headache	37 (9.34)
Asthenia	9 (2.27)
Hypoesthesia	5 (1.26)
Arthralgia	26 (6.57)
Injection site pain	71 (17.93)
Dyspnea	2 (0.51)
Myalgia	2 (0.51)
Rash	4 (1.01)
Fatigue	63 (15.91)
Pruritus	1 (0.25)

**Table 3 vaccines-11-00692-t003:** Laboratory results of participants with detectable anti-PF4 antibodies.

Laboratory Parameters	Participant with Detectable Anti-PF4 Antibodies (n = 12)
Hemoglobin (g/dL) (mean ± SD)	13.46 ± 1.17
White blood cell counts (×10^9^/L) (mean ± SD)	8.12 ± 1.93
Platelet counts (×10^9^/L) (mean ± SD)	296.50 ± 65.25
Platelet count < 150 × 10^9^/L [no.]	0
D-dimer (ng/mL) (mean ± SD)	288.58 ± 163.37
D-dimer > 500 ng/mL (no. (%))	1 (8.33)

**Table 4 vaccines-11-00692-t004:** Univariable analysis of the demographic and clinical parameters associated with positivity of anti-PF4 antibodies after receiving the first dose of the ChAdOx1 nCoV-19 vaccine.

Variables	Univariable Analysis		
	Odds Ratio	95% Confidence Interval	*p*-Value
Age	1.03	0.99–1.07	0.213
Female	0.51	0.16–1.56	0.246
Hypertension	3.04	0.95–9.78	0.062
Dyslipidemia	0.97	0.25–3.67	0.967
Diabetes mellitus	2.51	0.66–9.62	0.179
History of ischemic stroke	2.32	1.24–43.67	0.572
History of coronary artery disease	1.19	0.07–21.28	0.905
Anti-hypertensive drug	3.04	0.95–9.78	0.062
Anti-glycemic drug	2.87	0.75–11.03	0.125
Aspirin	0.89	0.05–15.75	0.939
Fever	1.78	0.55–5.73	0.334
Headache	0.88	0.11–7.00	0.903
Asthenia	1.58	0.09–28.71	0.757
Hypoesthesia	2.76	0.14–52.70	0.500
Arthralgia	0.54	0.03–9.39	0.673
Pain at injection site	3.44	1.06–11.18	0.040
Dyspnea	6.12	0.28–134.26	0.250
Myalgia	6.12	0.28–134.26	0.250
Rash	3.38	0.17–66.29	0.422
Fatigue	0.47	0.06–3.72	0.476
Pruritus	10.23	0.40–263.70	0.161

## Data Availability

All data pertinent to this study has been presented within the article.

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
