# Peer review of "Longitudinal Profiles of Anti-Platelet Factor 4 Antibodies in Thai People Who Received ChAdOx1 nCoV-19 Vaccination"

_vaccines, 2023, doi:10.3390/vaccines11030692_

Round 1

Reviewer 1 Report

The longitudinal study using ChAdOx1 nCoV-19 for detecting the anti-platelet factor 4 antibodies presents an exciting and well-designed work.

Authors need to make some modifications before it can be considered for publication.

In line 26, the authors need to modify this sentence “did not significantly change over time” it looks like something is missing, maybe ‘produce’ or maybe change significant to significantly.

The study was done on only the Thai population, so I suggest changing the title to Thai population instead of Asian, as Asians have several races which were not included in this study.

How could authors ensure the detected patients' availability for the second and third sample collections?

Figure 1 has not shown any dropouts, meaning all enrolled participants continued until 4 weeks after the first dose; it’s extremely difficult to manage.

The authors should also explain the time gap between the first and second doses of the vaccine, especially among those who had detectable antibodies.

Overall, I found only two patients who were different from those who had pre-vaccination detection of antibodies.

The authors should explain the range for them to consider a group weakly positive and how different they are from the positive group.

Figure 2 is not precise whether it’s a change in OD value or the average OD of each group.

Although rational approaches are required in disease management, merely based on only 2 samples where they found positive antibodies after vaccination, concluding that there is no need to test anti-PF4 antibodies when VITT is not suspected is a significant recommendation not supported by this study. Therefore, I propose a suitable correction to the statement.

Reviewer 2 Report

The authors need proper English editing for better readability and description.

Please highlight the results in the title. The current title is too broad. It may be "Vaccine- 14 induced immune thrombocytopenia and thrombosis associated anti-PF4 antibody is low in the Asian and did not significantly change over time after ChAdOx1 nCoV-19 vaccination".

Please double check the statistical analysis.
